# Genomic attributes of *Vibrio cholerae* O1 responsible for 2022 massive cholera outbreak in Bangladesh

Md Mamun Monir[1], Mohammad Tarequl Islam[1], Razib Mazumder[2], Dinesh Mondal[2], Kazi Sumaita Nahar[1], Marzia Sultana[1], Masatomo Morita [3], Makoto Ohnishi[3], Anwar Huq[4], Haruo Watanabe[3], Firdausi Qadri[1], Mustafizur Rahman[1], Nicholas Thomson [5,6], Kimberley Seed [7], Rita R. Colwell[4,8], Tahmeed Ahmed[9] & Munirul Alam [1] ✉

In 2022, one of its worst cholera outbreaks began in Bangladesh and the icddr,b Dhaka hospital treated more than 1300 patients and ca. 42,000 diarrheal cases from March-1 to April-10, 2022[1]. Here, we present genomic attributes of *V. cholerae* O1 responsible for the 2022 Dhaka outbreak and 960 7th pandemic El Tor (7PET) strains from 88 countries. Results show strains isolated during the Dhaka outbreak cluster with 7PET wave-3 global clade strains, but comprise subclade BD-1.2, for which the most recent common ancestor appears to be that responsible for recent endemic cholera in India. BD-1.2 strains are present in Bangladesh since 2016, but not establishing dominance over BD-2 lineage strains[2] until 2018 and predominantly associated with endemic cholera. In conclusion, the recent shift in lineage and genetic attributes, including serotype switching of BD-1.2 from Ogawa to Inaba, may explain the increasing number of cholera cases in Bangladesh.

*Vibrio cholerae*, native to the aquatic environment and the causative agent of cholera, has undergone continuous evolution in different parts of the world[3]. It gained mutations, genomic islands, and phages during its evolution[4]. According to a recent study, *V. cholerae* O1 El Tor strains responsible for the ongoing seventh pandemic evolved from non-pathogenic ancestors, acquiring the El Tor-form of the *tcpA* gene, CTX prophage, vibrio seventh pandemic island I (VSP-I), and vibrio seventh pandemic island II (VSP-II), and exhibited high spreading capability in 1961 in the Indonesian island, Sulawesi[5]. Strains of the El Tor biotype were isolated in Bangladesh in 1963, India in 1964, and in the former U.S.S.R., Iran, and Iraq from 1965 to 1966, and in Africa from 1970 to 1971[6]. The Ganges Delta of Bay of Bengal is the historical hotspot for the evolution of the *V. cholerae* pandemic clone[7]. According to a recent study, the 7th pandemic El Tor (7PET) strains spread out from the Bay of Bengal in at least three different but overlapping waves and there had been numerous transcontinental transmission events[8]. Genomic studies identified 13 transmission lineages (T1-T13) in Africa[9] and three transmission lineages (LAT1-LAT3) in Latin America[10]. And in recent years, two contemporary circulating lineages belonging to the 7th pandemic El Tor (7PET) wave-3 were reported in Asia[11]. Two recently circulating lineages in Bangladesh were identified as BD-1 and BD-2, differing significantly in genomic

[1]Infectious diseases division, icddr,b (International Centre for Diarrhoeal Disease Research, Bangladesh), Dhaka, Bangladesh. [2]Laboratory Sciences and Services Division, icddr,b (International Centre for Diarrhoeal Disease Research, Bangladesh), Dhaka, Bangladesh. [3]Department of Bacteriology, National Institute of Infectious Diseases (NIID), Tokyo, Japan. [4]Maryland Pathogen Research Institute, University of Maryland, College Park, MD, USA. [5]Parasites and Microbes Programme, Wellcome Sanger Institute, Wellcome Genome Campus, Hinxton, Cambridge CB10 1SA, UK. [6]London School of Hygiene and Tropical Medicine, LondonWC1E 7HTUnited Kingdom. [7]Department of Plant and Microbial Biology, University of California, Berkeley, CA, USA. [8]Johns Hopkins Bloomberg School of Public Health, Baltimore, MD, USA. [9]Nutrition and Clinical Services Division, icddr,b (International Centre for Diarrhoeal Disease Research, Bangladesh), Dhaka, Bangladesh. ✉e-mail: munirul@icddrb.org

attributes, e.g., mutant genes, heterogeneity in VSP-II, vibrio pathogenic island 1 (VPI-1), mobile genetic elements, toxin encoding elements, total gene abundance, phage-inducible chromosomal island-like element (PLE), and SXT-related integrating conjugative elements (SXT ICE)[2]. These genes and genomic islands have a crucial role in adaptation of the bacterium. For example, SXT ICE elements encode resistance to multiple antibiotics[12], VSP-II facilitates chemotactic responses and cell congregation[13], and PLE protects against bacteriophage infection[14]. The BD-2 strains had more SNPs and indels (insertion-deletions) than BD-1, and also richness in gene abundance, including antimicrobial resistance genes, gene cassettes, and PLE against bacteriophage infection, and were predominantly associated with endemic cholera in Bangladesh between 2013 and 2017[2].

Dhaka, the capital city of Bangladesh, experienced a massive cholera outbreak, the largest in 20–25 years. The icddr,b hospital treated a record high number of daily patients (more than 1300), numbering ca. 42,000 diarrhea cases between March 1 and April 10, 2022[1]. It was imperative to investigate isolates of the causative agent *V. cholerae*, employing whole genome sequencing and comparative genomics to understand the genetic drivers of such a massive cholera outbreak. Therefore, the genomes of 21 *V. cholerae* isolates from diarrhea samples collected between February and April, 2022, from patients admitted to the icddr,b hospital, Bangladesh, were sequenced and those sequences were compared with 267 genome sequences of strains from our laboratory collection (1991–2021), and 693 sequences retrieved from a public database[15], which included strains isolated from 1957 to 2017. In this work, we explore the recent evolutionary changes and transmission dynamics of the *V. cholerae* O1 El Tor Dhaka outbreak strains.

## Results

### Phylogenetic relationships and genetic clusters

The whole-genome sequence analysis of *V. cholerae* O1 El Tor strains associated with endemic cholera in Dhaka, Bangladesh, and Kolkata, India, showed two circulating lineages, one dominant in India and the other in Bangladesh[11]. To understand its evolutionary dynamics, a temporal genomic study of *V. cholerae* associated with endemic cholera in Dhaka, Bangladesh (1991–2017) was undertaken[2]. Results revealed distinct genomic attributes for the two circulating lineages, BD-1 and BD-2, which were negatively associated with each other in endemic cholera in Bangladesh. *V. cholerae* strains isolated between 1991 and 2022 in Bangladesh were the first to be analyzed and results showed the majority of strains isolated between 2018 and 2022 clustered with BD-1, formed a separate subclade (BD-1.2), indicating a shift back to predominance of BD-1 like strains in endemic cholera in Bangladesh (Fig. 1). *V. cholerae* strains phylogenetically related to BD-1 (Asian lineage 2) were predominant in India during the period when BD-2 strains were predominant in Bangladesh[11]. A comparative genomic analysis of Bangladeshi strains, including strains isolated in India between 1991 and 2016 showed BD-1.2 strains to be genetically similar to the Indian strains (Supplementary Fig. 1). For further investigation, a maximum-likelihood phylogenetic tree was constructed for a total of 981 genome sequences of *V. cholerae* O1 El Tor strains isolated from 88 countries between 1957 and 2022 (Supplementary Data 1). A total of 6399 high quality SNPs of non-repetitive, non-recombinant core genome sites were analyzed for maximum likelihood phylogenetic tree reconstruction. The revealed lineages of strains isolated in Bangladesh were annotated in the phylogenetic tree with strains of recently recognized transmission lineages (T9-T13) in Africa[9] and Latin American transmission 3 (LAT-3)[10]. Three distinct subclades of strains isolated in Bangladesh, BD-1, BD-1.1, and BD-1.2, clustered with a single clade of globally distributed strains (Fig. 2a). However, BD-2, representing the recent predominant lineage in Bangladesh, belonged to a separate clade that included strains isolated in Asia. Henceforth, the first clade is here referred to as the global clade and the clade containing BD-2 strains as the Asian clade. Strains of the global clade have been reported to be associated with endemic and epidemic cholera in different parts of the world, including the 2010 cholera epidemic in Haiti[16] and the 2016–2017 in cholera epidemic in Yemen[17]. In the constructed phylogenetic tree, BD-1 strains isolated during 1999–2007

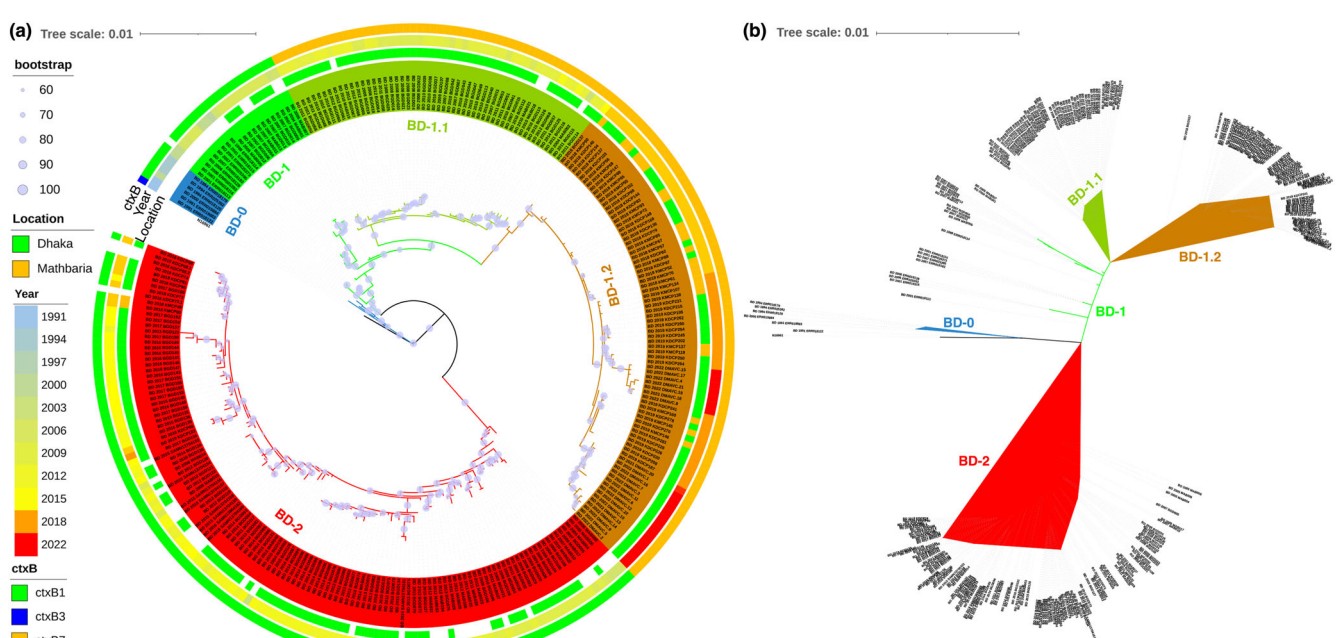

**Fig. 1 | Phylogenetic clustering of Bangladesh *V. cholerae* strains. a** Maximum likelihood phylogenetic tree rooted from out-group reference strain *Vibrio cholerae* N16961 based on whole genome SNPs and the strains belonging to lineages BD-0, BD-1, BD-1.1, BD-1.2, and BD-2. Metadata of isolates are displayed in rings according to color scheme on the left. Lineages BD-0, BD-1, BD-1.1, BD-1.2, and BD-2 are defined by blue, light green, green, orange, and red tree branches, respectively. The majority of strains isolated between 2018 and 2022 clustered with BD-1.2. **b** Unrooted tree showing the difference between the two subclades BD-1.1 and BD-1.2 of the clade BD-1. Genetic variant data used are provided in Source Data 1 and related metadata in Supplementary Data 1.

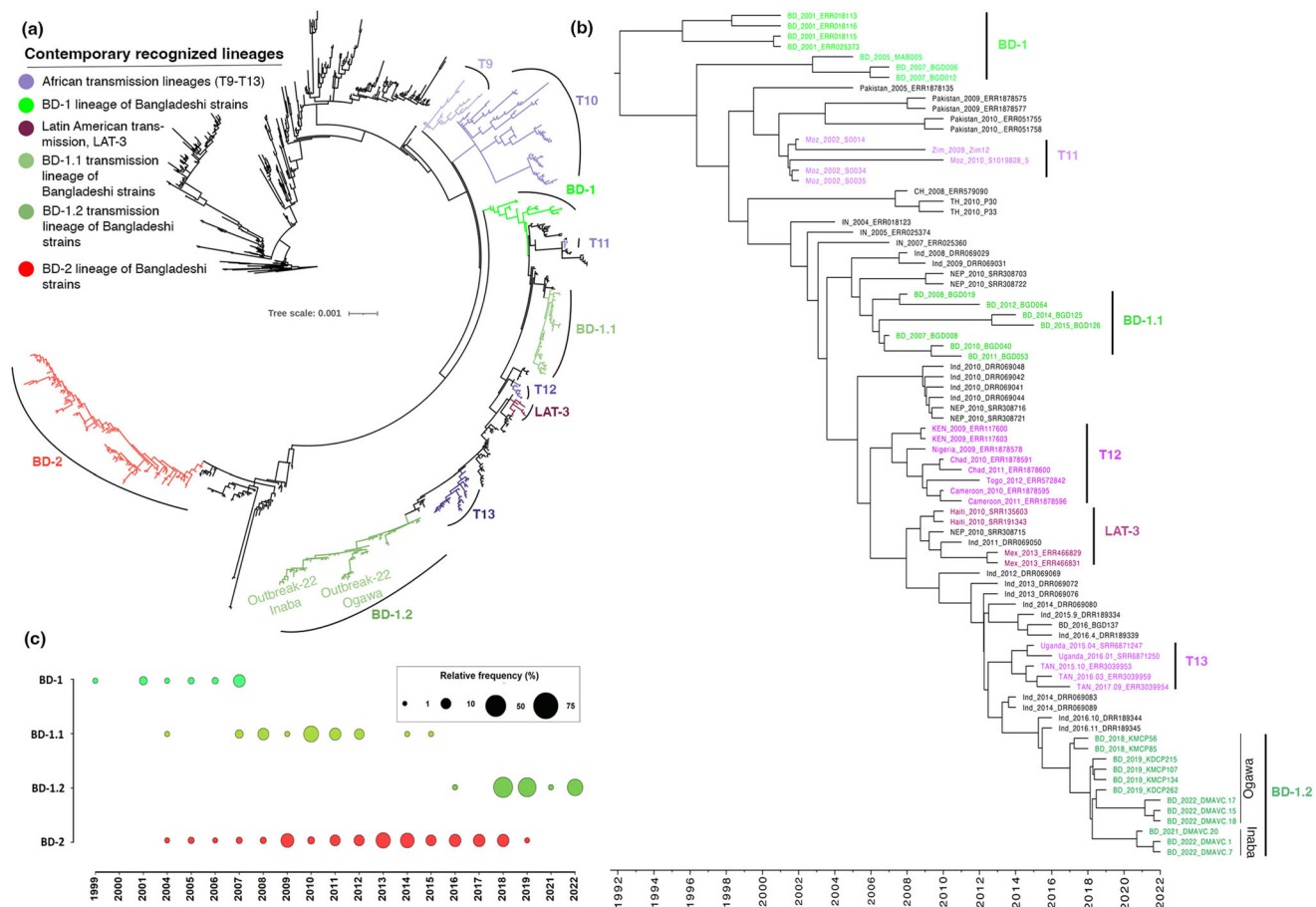

**Fig. 2 | Phylogenetic relatedness of *V. cholerae* O1 El Tor stains isolated during the 2022 cholera outbreak in Dhaka, Bangladesh. a** Maximum likelihood phylogenetic tree based on 981 genome sequences. Outgroup is represented by a strain isolated from Indonesia in 1957. Recently recognized and newly identified lineages of strains were annotated in the tree. Clades of isolates are colored according to the color scheme at the left. Lineages BD-1, BD-1.1, BD-1.2 are defined using color green of differing intensity. BD-2 is defined by the red color of the tree branches. All transmission lineages in Africa are colored purple with differing intensity, Latin American Transmission 3 lineage is represented in dark magenta. **b** BEAST analysis of representative strains of the clade, where BD-1, BD-1.1, and BD-1.2 clustered with the strains isolated in different countries. **c** Relative frequencies of strains per year. Size of the circles shows relative frequency of sequenced strains belonging to each clade in the scheme. Genetic variant data used for phylogenetic and phylodynamic analyses are provided in Source Data 1, and related metadata in Supplementary Data 1. Frequencies of strains for the lineages are provided as Source Data 2.

located at the bottom of the global clade (Fig. 2a), followed by strains from China, Thailand, Mozambique, Zimbabwe, India, and Nepal. BD-1.1 strains isolated from Bangladesh between 2008 and 2014 clustered below the Indian strains isolated between 2008 and 2009. BD-1.1 strains differed from BD-1 strains in certain genomic features, namely BD-1 strains had *ctxB1* genotype, while BD-1.1 strains had *ctxB7* genotype, similar to strains isolated from India and Nepal. Strains carrying the *ctxB7* genotype were reported in the massive Haitian cholera outbreak in 2010[16]. BD-1.1 subclade strains overlapped in predominance with BD-2 strains in endemic cholera in Bangladesh. However, strains of the subclade have not been found clinically since 2015.

African strains (T12) formed a distinct subclade in the global clade, closely related to strains isolated in India and Nepal in 2010, followed by Latin American (LAT-3) strains, reported as having been from Africa in the earlier study[10]. Another report suggested relatedness of strains in Africa (T13) with Indian strains isolated between 2011 and 2015[9]. Strains isolated between 2018 and 2022 in Bangladesh formed a subclade, BD-1.2, separate from T13 strains and phylogenetically close to strains isolated from India in 2016. Hence, the Bangladesh outbreak strains may be an expansion of the global clade as a subclade BD-1.2 (Fig. 2a, b).

Results of Monte Carlo Markov Chain (MCMC) phylodynamic analysis using BEAST[18], showed BD-1.1 and BD-1.2 strains from the present study with their recent common ancestor (MRCA) to be cholera strains isolated in India in 2005 (95% height posterior density (HPD): 2004-2006) and 2015 (95% HPD: 2015-2016), respectively (Fig. 2b). The data suggest temporal shifts in predominance among the clade-specific strains isolated from Bangladesh (Fig. 2c). BD-1 and BD-2 overlap but negatively related in predominance from 2001 to 2007. From 2007 to 2015, strains of BD-1.1 subclade and BD-2 were detected. However, in subsequent years, BD-2 comprised most of the isolates from 2013 to 2017. Interestingly, BD-1.2 strains were isolated in high numbers in 2018 continuing in subsequent years, suggesting a reverse shift to global clade in Bangladesh. The results suggest endemic and epidemic cholera in Bangladesh had progressed via evolution, transmission, and temporal transition to predominance between lineages over the years.

### Emergence of the contemporary lineages
Pandemic *V. cholerae* strains continue to evolve by acquiring mutations in their core genome, thus introducing various lineages. In addition to the six defined lineages reported in prior studies[2,9,10], nine additional lineages/groups of global and Asian clades were defined (Supplementary Data 2 and Fig. 3a). These include BD-1.1 and BD-1.2 in Bangladesh, IND-1, IND-1.1, IND-1.2, IND-1.3, and IND-2 from India, and AS-1 and AS-2, isolated from several Asian countries. SNPs and indels

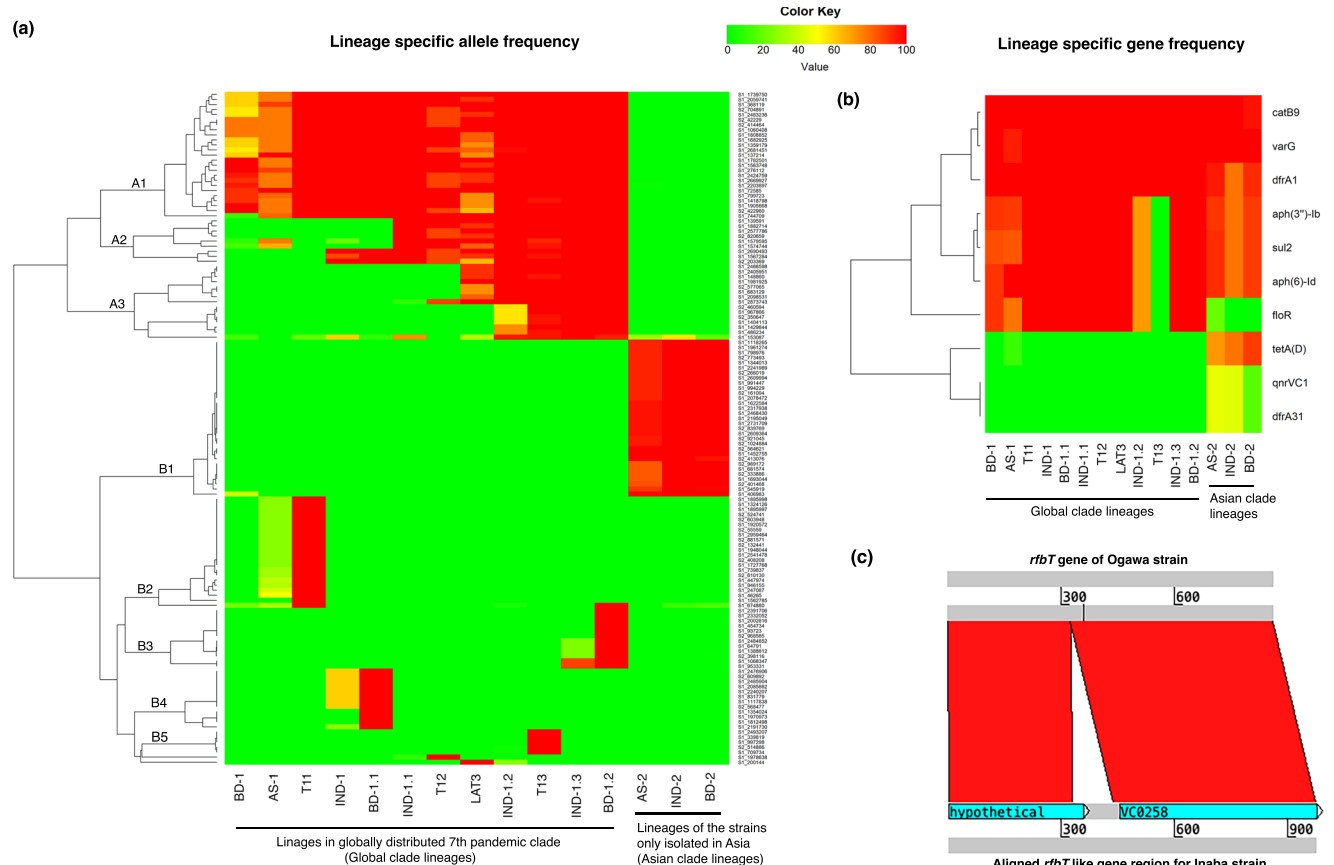

**Fig. 3 | SNP clusters and AMR genes associated with lineages, and genetic changes in serotype switching. a** Highly significant SNPs identified using chi-square test with two tailed p-values and plotted using heatmap by clustering SNPs. *X*-axis shows the lineages/groups according to chronological order in the phylogenetic tree, left *y*-axis showing SNP clusters and right *y*-axis ID of the lineage-associated SNPs. Frequency of the alternative allele of each lineage/group were plotted. BD-1, BD-1.1, BD-1.2, and BD-2 are lineages/groups of strains isolated from Bangladesh. IND-1, IND-1.2, IND-1.3, and IND-2 are lineages/groups of strains isolated from India. AS-1 and AS-2 are groups of the strains isolated from more than one Asian country. T11, T12, and T13 are African transmission lineages. LAT-3 is the Latin American transmission lineage. **b** Frequency of antimicrobial resistance genes in the different lineages. Same color scale was used for heatmap as in fig-2a. And, (**c**) *rfbT* gene sequence comparison between Ogawa and serotype switched Inaba strains of BD-1.2. Source data for fig-3a are tabulated in Supplementary Data 2, and for fig-3b in Supplementary Table 4. And, sequence data used for fig-3c are provided as Source Data 3.

(insertion-deletions) of the lineages were compared to identify the set of mutations acquired during emergence of the lineages and genetic changes with adaptive implications. Chi-square test was used to analyze allelic diversity among the lineages to identify associated SNPs and indels (Fig. 3a). A total of 134 SNPs had alternative alleles in all strains of at least one lineage. Hierarchical cluster analysis showed several SNP groups in one of the emergent parental lineages and sustained in descendent lineages. For example, the accumulation of SNPs in cluster A was observed in strains of lineages of globally distributed 7th pandemic El Tor clade. A total of 26 SNPs identified in subcluster A1 were common in strains of the global clade (Supplementary Data 2). Asian clade lineages had wild type SNP alleles of El Tor reference strain N16961. Similarly, 31 SNPs of subcluster B1 were detected in all lineages of the Asian clade, but not in the global clade. SNPs of subclusters A2 and A3 were accumulated by a specific lineage of the global clade, e.g., 9 SNPs were in A2 cluster of which two SNPs accumulated by AS-1, three by IND-1, and four by IND-1.1. Again, 15 SNPs in cluster A3 were acquired by different lineages of the global clade. SNP clusters B2, B3, and B4 suggest recent transmission events, based on allele sharing of SNPs. Asian strains (AS-1) appear to be recent ancestor of an African transmission lineage (T11) based on SNP cluster B2, Indian strains (IND-1) as a recent ancestor of BD-1.1 based on B4, and IND-1.3 as a recent ancestor of BD-1.2 based on B3 (Fig. 3). Gene enrichment analysis was conducted for genes corresponding to lineage-associated SNPs (Supplementary Figs. 2–4). According to gene ontology (GO)

results, the genes showing mutations in the outbreak strains were involved in important biological processes, such as cell wall organization, cell-to-cell communication, toxin transport, modulation process of yet another organism, importing into cell, DNA integration, and chemotaxis that constitute a complex network. Such complex networks of molecular functions, including toxin transmembrane transporter activity, porin function, may have contributed to environmental fitness and infection potential of the bacterium.

## Differences among BD-1.1, BD-1.2, and BD-2 strains

BD-1.1 is a recent and BD-1.2 the current Bangladesh lineage of strains, with respect to the global clade. Phylogenetic analysis and SNP-clustering provide compelling evidence for BD-1.2 strains having appeared in Bangladesh recently, rather than having derived locally from BD-1.1, which displaced locally dominant BD-2 strains responsible for endemic cholera until recently. The BD-1.2 strains responsible for the 2022 Dhaka outbreak differed from BD-2 in major genetic characteristics, antimicrobial resistance, and epidemiological behavior (Table 1). For example, BD-1.2 strains harbored *ctxB7* genotype while BD-2 strains had *ctxB1*. Phage-inducible chromosomal island-like elements (PLE), which protect bacteria from bacteriophage infection, found in most of the BD-2 strains, but not in the majority of BD-1.2 strains. BD-1.2 and BD-2 strains had different types of SXT ICE element, VSP-II, VPI-1, and *gryA* gene alleles. Most importantly, BD-2 strains clustered in a single expanded clade, namely Asian clade, with strains isolated from Asian

**Table 1 | Major differences between the recently recognized BD-1.2 and formerly predominant BD-2 strains**

| Attributes | BD-2 | BD-1.2 |
|---|---|---|
| *ctxB* allele | B1 | B7 |
| Type of SXT/ICE | Except for a few, mostly ICE$^{TET}$ | Mostly similar to ICE$^{GEN}$/ICEVchInd5/ICEVchBan5 |
| VSP-II | var. 2 ($\Delta VC\_495$-$VC\_500$) or var. 3 ($\Delta VC\_491$-$VC\_501$) | var. 4 ($\Delta VC\_495$-$VC\_512$) |
| VPI-1 | variant ($VC\_819$-$\Delta VC\_821$) | wild type |
| gryA gene allele | haitian *gryA* ser83 to Ile, asp660 to glu | haitian *gryA* ser83 to Ile |
| Genetic similarity | cluster in a single expanded clade with Asian strains | cluster in single expanded clade with globally isolated strains including T11-T13, LAT-3, BD-1, and BD-1.1 |
| Epidemiology | associated with endemic cholera in Bangladesh between 2004 and 2019, and predominantly during 2013–2017 | newly recognized and associated with 2022 massive cholera outbreak in Bangladesh |

Here, B1 and B7 refer to ctxB1 and ctxB7, respectively. Phage-inducible chromosomal island-like elements (PLE), which protect bacteria from bacteriophage infection, found in most of the BD-2 strains, but not found in majority of BD-1.2 strains. BD-2 and BD-1.2 strains had different types of SXT ICE element, VSP-II, VPI-1, and gryA gene allele. BD-2 strains cluster in a single expanded clade, namely Asian clade, with the strains isolated from Asian regions, whereas BD-1.2 strains cluster in a single expanded clade, namely global clade, with the strains isolated from different parts of the world. BD-2 strains were highly predominant between 2013 and 2017, and BD-1.2 recently introduced and associated with the massive outbreak 2022 in Bangladesh.

countries and associated with endemic cholera in this region[2,11]. BD-1.2 strains clustered in a single expanded clade, namely global clade, with strains isolated from different parts of the world and associated with endemic and epidemic cholera in recent years[9,10,17,19].

Although both BD-1.2 and BD-1.1 strains belong to a same extended global clade, BD-1.2 strains have a number of novel mutations absent in the BD-1.1 strains. Several missense SNP mutations, with different alleles, were detected in BD-1.2 strains when compared with BD-1.1 strains (Supplementary Table 1). Of these SNPs, 21 had reference alleles for BD-1.1, but alternative alleles for BD-1.2 strains. A new missense SNP mutation resulting in Arg491His in the penicillin-binding protein 3 (PBP3) domain (267-562) of the gene *VC_2407* was detected in all BD-1.2 strains. A recent study showed PBP3 is essential for growth of *Pseudomonas aeruginosa*[20]. It is interesting to suggest the mutation in PBP3 may accelerate growth and adaptability of BD-1.2 strains, which also have an altered protein structure of the bile salt resistance genes *ompU*[21] that forms passive diffusion pores to allow small molecular weight hydrophilic materials to move across the outer membrane (EggNOG: COG3203). Bile salt disorganizes cell membrane structure and also triggers DNA damage[22]. A mutation in membrane-bound lytic murein transglycosylase D precursor gene *mltD* was detected in for all BD-1.2 strains and is involved in the peptidoglycan metabolic process (Uniport ID: Q9KPX5). *E. coli mltD* is involved in cell wall organization and plays a role in recycling muropeptides during cell elongation and/or cell division (UniPort ID: P0AEZ7). A mutation in this virulence-related gene in *Vibrio anguillarum* has been shown to enhance lethality in zebrafish[23]. BD-1.2 strains had a mutation in the *skp* gene encoding a protein, a molecular chaperone of gram-negative bacteria, required for formation of soluble periplasmic intermediates of outer membrane proteins[24]. BD-1.2 strains also had a missense mutation (Ala582Gly) in multifunctional-autoprocessing repeats-in-toxin holotoxin (MARTX) *rtxA* gene. All BD-1.1 strains had alternative alleles for 10 SNPs while the BD-1.2 strains had reference alleles for the SNPs. Along with missense SNPs, allelic differences in several synonymous and intergenic SNPs and indels were observed between BD-1.1 and BD-1.2 strains (Supplementary Tables 2 and 3). A total of nine synonymous SNPs and five intergenic SNPs had different alleles in BD-1.1 and BD-1.2 strains. In addition, eight indels revealed different alleles for strains of these subclades.

### Antimicrobial resistance (AMR) and related phenotypes

AMR genes of strains comprising the global and Asian clades were studied using the bioinformatics pipeline ABRicate[25]. Results of the genome analysis revealed essentially a uniform pattern for drug resistance in lineages of the global clade, including 2022 Bangladesh outbreak strains. African transmission lineage T13 showed three AMR genes, *varG*, *catB9*, and *dfrA1*, are associated with resistance to

carbapenem, chloramphenicol, and trimethoprim, respectively (Supplementary Table 4). All other lineages of the global clade had four additional genes, with *aph(6)-Id* and *aph(3″)-Ib* associated with streptomycin, *sul2* with sulfonamide, and *floR* with chloramphenicol and florfenicol resistance. Lineages of the Asian clade revealed a different pattern for drug resistance compared to strains of the global clade. AMR gene *floR* was detected in 20% of Asian clade AS-2 strains, but not in descendent lineages IND-2 and BD-2. Tetracycline resistance *tetA*(D) was detected in 71–89% of the strains and trimethoprim resistance gene *dfrA31and* quinolone resistance *qnrVC1* in 17–46% of Asian clade lineage strains. The AMR gene profile suggests BD-1.2 is an expansion of the global clade in Bangladesh, differing from T13. In addition to AMR some strains carried additional genes contributing to multidrug-resistance (MDR) and potentially highly drug-resistant (XDR). An XDR strain of BD-2 lineage was isolated from a clinical specimen in Dhaka during 2019. The strain harbored 14 AMR genes and was resistant to all tested drugs available in our laboratory at the time. Emergence of the BD-2 strain exhibiting MDR/XDR occurred when strains of the lineage had essentially been superseded by BD-1.2, associated with cholera in 2019. Although the MDR/XDR was not subsequently detected, the data suggest that BD-2 strain acquired resistance genes due to intense selective pressure at that time to combat the widely circulating BD-1.2. Nucleotide blast was used to match contigs of representative strains of Asian and Global clades with seven publicly available sequences of the Integrative and conjugative elements (ICEs)- ICEVchban5 (GQ463140.1), ICEVchind4 (GQ463141.1), ICEVchind5 (GQ463142.1), ICEVchmex1 (GQ463143.1), ICEVflInd1 (GQ463144.1), ICE$^{TET}$ (MK165649.1), and ICE$^{GEN}$ (MK165650.1). Except for a few strains, blast searches for global clade strains produced high bit scores when aligned with ICE$^{GEN}$ (MK165650.1), ICEVchInd5 (GQ463142.1), or ICEVchBan5 (GQ463140.1). In contrast, for the majority of Asian clade strains, the high bit score was obtained when aligned with ICE$^{TET}$ (MK165649.1) (Supplementary Data 3) and, in addition, mutation Ser 83 Ile in *gryA* commonly found in the Global and Asian clade strains. However, the BD-2 strains isolated since 2009 contained an additional mutation (Asp 660 Glu). All strains of the global and Asian clades had a mutation in *ParC* (Ser 85 Leu), with the exception of a small number of BD-1 ($n = 10$) and AS-2 ($n = 2$) strains.

### Serotype switching

Two BD-1.2 serotypes, Ogawa and Inaba, were encountered and temporal analysis results revealed all BD-1.2 strains isolated between 2016 and 2019 were Ogawa serotype[26], whereas the serotype predominantly associated with recent cholera outbreaks in Yemen, Tanzania, and Uganda was Ogawa[17]. The Bangladesh clinical data indicated BD-1.2 serotype switching from Ogawa to Inaba, the serotype that was prevalent as the causative agent of cholera since September, 2020 (Supplementary Table 5).

Serotype conversion from Ogawa to Inaba has been linked to mutations in the *rfbT* gene[22,23] as it has also been shown that the gene coding for serotype can be laterally transferred[5]. To identify the genetic basis of the recent serotype switch in Bangladesh, sequences of *rfbT* were extracted from both Inaba and Ogawa serotype strains. Comparison of the sequences showed all of the Inaba strains had an insertion (nt position: 27–28; ins113bp for 2 strains and ins112bp for 12 strains) within *rfbT* gene (Supplementary Figs. 5 and 3c).

### Determination of *ctxB* allele and drug response pattern of additional strains

We performed comparative genomic analysis of 21 *V. cholerae* O1 El Tor strains isolated from the peak of the 2022 cholera outbreak to show their phylogenetic relationship with 960 whole genome sequences retrieved from the public database. Our results suggest a recent shift in the predominance of *V. cholerae* strains responsible for endemic cholera in Bangladesh. Results also suggest that the 2022 massive cholera outbreak was attributed to BD-1.2 strains that were successful in replacing the BD-2 strains predominantly associated with endemic cholera in Bangladesh. We tested additional *V. cholerae* by randomly selecting 30 strains isolated between March and September 2022 in order to support our study findings and strengthen the conclusions made. The *ctxB* genotype was determined by a double-mismatch-amplification mutation assay (DMAMA) PCR and drug response patterns using disk diffusion assays[27]. According to our results, all of the tested strains carried *ctxB7*, and proved tetracycline sensitive as observed for the BD-1.2 (Supplementary Table 6). These results appeared in sharp contrary to the BD-2, which carried *ctxB1*, and tetracycline resistance as markers, supporting the overall findings of the present study.

## Discussion

The Ganges Delta of the Bay of Bengal, Bangladesh, historically is considered an ancestral home of cholera with the disease endemic there for centuries[28,29]. In this study, we investigated genomic attributes of *V. cholerae* O1 El Tor associated with the 2022 massive cholera outbreak in Dhaka, Bangladesh. A genomic analysis of outbreak strains isolated from admitted patients at icddr,b hospital was done, including comparative genomic analysis of sequence data of earlier studies[2,9–11] in 88 countries from 1957 to 2022. The results provided evidence for 7th pandemic El Tor wave-3 global clade strains forming a new subclade, designated BD-1.2, tracing to a most recent common ancestor, namely a globally distributed 7th pandemic El Tor lineage predominantly associated with recent endemic cholera in India. Cholera is well established as a climate-driven disease[30] with *V. cholerae* resident flora in aquatic environments[31]. Our results indicate an earlier presence of strains belonging to the subclade BD-1.1 in Bangladesh around 2007. Both, the resident BD-2, predominantly associated with cholera in Bangladesh, and the new BD-1.1 strains showed a negative relationship in predominance, the latter not associated with any major outbreak before last isolated in 2015. Emergence of BD-1.2 strains in Bangladesh was striking thereafter. Both BD-1.1 and BD-1.2 comprise a component of a global clade that includes strains associated with cholera world-wide, e.g., Haiti[16] and Yemen[17]. The Bangladesh BD-1.2 strains initiated a massive cholera outbreak in 2022 displacing BD-2 strains locally dominant at the time[2]. Intense niche competition with BD-2 strains may account for this turn of events. In addition, the BD-1.2 strains proved to be genetically different from recently recognized transmission lineages T11-13 and LAT-3[9,10] and circulating lineages in Bangladesh[2]. They carried unique mutations in several genes playing a critical role in promoting growth, resistance to bile salt, cell wall organization, and toxigenicity, presumably selectively advantageous for this subclade. While most of the global clade strains are known to possess seven AMR genes, T13 strains harbored only three, similarly observed for

30% of related Indian strains, hence some evidence for concluding the relatively drug-sensitive T13 strains transmitted from India to Africa as the T13 strains. A BD-2 strain carrying 14 AMR genes (*aac(6')-lb-cr5, aph(3")-lb, aph(6)-ld, blaOXA-1, blaPER-3, catB3, catB9, dfrA1, dfrA15, mph(A), sul1, sul2, tetA(D)*, and *varG*) and showing resistance to all of the eleven tested drugs (Cephalothin (KF), Streptomycin (S), Cefixime (CFM), Ceftriaxone (CRO), Nalidixic Acid (NA), Sulfamethoxazole/Trimethoprim (SXT), Cefepime (FEP), Mecillinam (MEL), Ciprofloxacin (CIP), Ampicillin (AMP), Aztreonam (ATM))[27] may be considered a remnant of the lineage during transition while BD-1.2 was predominant, with perhaps extreme drug resistance the result of intense selective pressure as the two lineages fights to establish their respective niches[32]. The recent observation of the switch in serotype from Ogawa to Inaba of BD-1.2, coexistence of both serotypes, and subsequent massive cholera outbreak in Bangladesh is remarkable, since cholera outbreaks have been linked with serotype switching[33]. Ogawa and Inaba serotypes do not appear to differ in severity or duration of illness caused[33,34]. However, Ogawa serotype offers less protective immunity than Inaba from reinfection with the heterologous serotype[34]. Thus, it can be concluded that observed genetic changes including serotype switching from Ogawa to Inaba may be associated with the high number of cases that occurred during the 2022 Dhaka outbreak. Although BD-1.2 strains comprise a component of the global clade, its notoriety as a devastating pathogen is demonstrated very clearly by the extent of the outbreak with which it is linked, humbling locally dominant BD-2 strains and reflecting greater risk this new subclade may cause even more devastating epidemics.

## Methods

### V. cholerae strains

*Vibrio cholerae* O1 El Tor strains used in the present study were isolated from dipsticks positive stool samples collected from patients admitted at icddr,b hospital during the 2022 massive cholera outbreak. Stool sample were collected according to the study protocol reviewed and approved by icddr,b institutional review board (Research review and ethical review committees). The 21 outbreak strains subjected to whole genome sequencing in the present study were compared with 960 El Tor strains of the ongoing 7th cholera pandemic isolated between 1957 and 2021 from 88 countries. Additional 30 El Tor strains isolated from cholera patients admitted at icddr,b hospital (March–September, 2022) were tested for serotype, *ctxB* genotype, and drug resistance patterns[27] to increase the power of the study. This study did not need participant consent since there was no risk involved, no personally identifiable information or identifiable biospecimens were collected, and no follow-up was made after collecting the stool samples.

### Whole genome sequencing

Genomic DNA was extracted from pure broth culture of *V. cholerae* isolates using Qiagen DNA Extraction Kit as per the manufacturer's instructions. DNA QC and quantification were performed employing a NanoDrop 1000 spectrophotometer (Thermo Fisher Scientific, United States) and Qubit 4.0 fluorometer (Life Technologies). The whole-genome sequencing libraries were prepared from 300 to 350 ng of genomic DNA using Illumina DNA Prep Library Preparation Kit (Illumina) according to the manufacturer's instructions. Bacterial whole genome sequencing was carried out at the icddr,b Genomics Center of the International Center for Diarrheal Disease Research, Bangladesh (icddr,b). The 150 bp paired-end sequencing reads were generated on an Illumina NextSeq 500 system using a NextSeq Mid output v2.5 reagent kit.

### Bioinformatics analysis

Initially, fastp v0.23.2[35], an ultra-fast quality control analysis program that inspects raw paired-end reads and filters out faulty ligation or

adapter sections, was used to evaluate the quality of the raw shotgun paired-end sequences. Spades v3.15.4[36] and ragout v2.3[37] genome assemblers were used to generate contigs and reference-based scaffolds, respectively. Prokka v1.14.5[38], a bacterial genome annotation tool, was used to annotate the whole genome. The antimicrobial resistance gene profiles for all strains were determined using ResFinder[39] and ABRicate v1.0.1[25]. Phylogenetic analysis was conducted using IQ-TREE v2.2.0[40] with 1000 bootstrap and best fitted evolutionary model selected using ModelFinder[41]. Phylodynamic analysis was carried out using BEAST v.2.6.7[18] according to the parameter setting described in a prior study[9]. Reference sequences JX565645-JX565687 were used as query and scaffolds of strains as subjects in blast searching for extracting *rfbT* gene sequences. Functional gene enrichment analysis was conducted using PANNZER[42] and GSEA-Pro.v3 (http://gseapro.molgenrug.nl), and network of GO terms were constructed using REVIGO[43]. Other bioinformatic analyses used in this study were described in supplementary methods.

## Statistical analysis
Genome wide association study using Pearson's chi-squared test was conducted to identify lineage/group associated SNPs and indels (degrees of freedom = 14). Contingency tables for each of tested SNPs were constructed with respect to lineages for testing significance of association between lineages and SNPs. In house R-script was used for the analysis.

## Reporting summary
Further information on research design is available in the Nature Portfolio Reporting Summary linked to this article.

## Data availability
Newly sequenced data used in this study were submitted under Bioproject accessions IDs PRJDB13928 and PRJDB13857. Publicly available sequence data used in this study downloaded from the European Nucleotide Archive (ENA), and metadata along with accession numbers are provided in Supplementary Data S1. In addition, other relevant data used are provided in supplementary data, and supplementary tables. Source data are provided with this paper.

## Code availability
The source code for the analysis performed for this study is available on Github (https://github.com/mamunmonir/Vibrio_genomics)[44].

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

## Acknowledgements

This work was supported in part by icddr,b, National Institutes of Infectious Diseases (NIID), Tokyo, the Research Program on Emerging and Re-emerging Infectious Diseases (JP21fk0108139) from the Japan Agency for Medical Research and Development (AMED), the National Institute of Allergy and Infectious Diseases (NIAID), the Foreign, Commonwealth and Development Office (FCDO)/Wellcome), the National Science Foundation (NSF) and the National Institutes of Health (NIH). M.A. was supported by AMED, NIAID (R01AI039129) and FCDO)/Wellcome (215704/Z/19/Z), R.R.C. was by NSF (OCE1839171 and CCF1918749), and M.A. and K.S. by NIH (R01AI53303). Authors acknowledge icddr,b hospital and laboratory staff for their support. icddr,b gratefully acknowledges the following donors for providing unrestricted support: Governments of the People's Republic of Bangladesh, Global Affairs Canada (GAC), Swedish International Development Cooperation Agency (Sida), and the Foreign Commonwealth & Development Office (FCDO), UK. All the authors read and approved the final manuscript.

## Author contributions

M.A. and M.M.M. designed this study. M.T.I., K.S.N., and M.S. collected stool samples from hospitalized patients, isolated strains, cultured, and performed DNA extraction. R.M., D.M., and M.R. sequenced outbreak strains. M.M.M. performed data analyses and written draft manuscript. R.R.C., T.A., K.S., N.T., F.Q., H.W., A.H., M.O., M.M., M.T.I., and M.A. edited the draft Manuscript. M.A. supervised this study, and all authors contributed to the manuscript.

## Competing interests

The authors declare no competing interests.
