## [Peer review file · Nature Communications]

REVIEWER COMMENTS

Reviewer #1 (Remarks to the Author):

The authors investigated the genomic characteristics of *V. cholerae* O1 responsible for the 2022 Dhaka cholera epidemic in a global phylogenetic context of 960 7PET strains from 88 countries. They found that the 2022 outbreak was caused by a subclade, BD-1.2 that was present in Bangladesh since 2016. Moreover, they claimed that this subclade successfully established dominance over the endemic BD-2 clade mainly through genetic changes including serotype switch in BD-1.2 from Ogawa to Inaba.

Specific comments

1. In the title add O1 to *Vibrio cholerae*
2. Since BD 1.2 has existed since 2016, it should not be referred to as a “new” subclade.
3. Since this manuscript might be coming out when the outbreak is over, I suggest that sentences such as “Bangladesh, currently is experiencing massive cholera” be revised to reflect this was in the recent past.
4. The study is essentially reporting on the recent 2022 outbreak, however, only 21 isolates from this outbreak were included in the analyses. That is quite a limitation to have 21 isolates while there were about 40,000 diarrhea cases within a month. This has to be addressed because it can have a significant effect on the conclusions of the study, especially about the subclade BD1.2 found in these 21 isolates, knowing that a larger strains collection for this huge outbreak could potentially cover and reveal more subclades...
5. For information, it would be good if the authors provide in bracket the years of isolates of the 267 genome sequences added from their laboratory collections, as this helps to understand their epidemiological link to the current outbreak. For instance, Fig 2 C shows a population replacement/takeover by BD1.2. This interpretation is however based on just 21 isolates... in other words, we cannot rule out here that BD2 strains were not part of the 2022 outbreak.
6. In fig 1a, the title/names of each ring are needed and should be readable as this will guide the reader to know which ring corresponds to which legend.
7. There is a need for a better reference announcing the 2022 outbreak compared to the press release in Reference 1... it would be of interest to see if the outbreak has ended and how many total cumulative cases have been recorded now.

8. I'm in doubt whether strain DMAVC-20 from Nov 21 was part of this outbreak (in Table S1, it is also described as "this study" in the "reference column"

9. The extended data in figure 2c is complex to interpret and might be useful to add more legends

10. The Yemen and Tanzanian T13 strains have ctxB7 and are Ogawa serotypes with MLST 69 and reported to originate from Asia. However, the BD1.2 strains in this study which seem to be related to the African T13 strains are rather Inaba. Could the authors please confirm the MLST of these BD1.2? For information, most VC O1 Inaba tend to be of ST515 and often with a ctxB1, which is not the case here.

11. Can the authors also please show the mutations in the wbeT region that are known to be responsible for the switch from Ogawa to Inaba (often Del_TGTAC at nt position 24,28)? Because figure 3c from Artemis (about rfbT) is not informative enough on this.

12. It would be important to know and compare how was the ICE SXT in the BD-1.2 strains when describing the AMR data. Similarly, changes in the quinolone resistance genes parC and gyrA are important markers in the resistance epidemiology of V cholerae.

13. The method is well described and the study well conducted, a suggestion is that RaxML generates better probability compared to IQ-Tree.

14. As one of the key take-home messages from this paper is the takeover of BD1.2, a table included directly in the manuscript showing characteristic differences between BD1.2 and BD2 should be added.

15. If possible from archives, authors are encouraged to enrich the tree with more strains from the 2022 outbreak to increase the power of the conclusions.

Reviewer #2 (Remarks to the Author):

This manuscript reports on the genetic characteristics of the 2022 cholera outbreak in Bangladesh, based on sequences from 21 clinical isolates. It compares these genomes with ~960 sequences retrieved from public databases as well as from their laboratory collection. The data indicate that the strains belonged to a separate subclade of BD-1-like strains, which represented a switch from the previously dominant BD-2 lineage. The data presented here are timely and noteworthy, and once again, demonstrates how genomic microbial surveillance of clinically sourced isolates can provide important insights into the spread of disease and evolution of virulence at local and global levels.

The manuscript contains appropriate descriptions of the whole genome sequence data and the comparative analyses performed are consistent with the methodology used in this field.

Comments for the authors:

1. The introduction to the manuscript is somewhat challenging to follow for a non-Vibrio expert:

a. Line 72 (and elsewhere e.g. line 81)– these are Vibrio specific terms that should be defined and their importance explained.

b. Line 77-79+ - As written it is not clear if the lineages mentioned in the first sentence are the same as or distinct from those mentioned in the second sentence. The manuscript needs a better introduction to global Vibrio phylogeny and then the local situation in Bangladesh.

c. Line 89 – poor phrasing “It was concluded imperative”

2. Number of strains sequenced – the authors sequenced quite a small number of isolates (n=21) in relation to the outbreak size (~40,000 cases a month) and when compared to the numbers sequenced in other (similar) cholera papers. Why was this number chosen and how were these selected from the huge numbers of patients admitted to hospital (e.g. patient or disease characteristics?). Is this number sufficient to describe the outbreak agent?

3. Line 294-6 – check this sentence as it does not make sense to me “...yet victims infected with either serotype generally not develop clinically apparent disease”. Aren’t strains all taken from hospitalised patients?

Figure related queries:

Figure 1 – what is significance of including/distinguishing Dhaka vs Mathbaria locations for the phylogenetic tree?

Figure 2c – this is a nice pictorial representation of the strain changes over time. Is this data specific to strains isolated only in Dhaka? If so, the image infers that between 10-50% were found to belong to BD-1.2 in 2022 - where do the remaining strains situate in terms of their genetics? How does this fit with the authors conclusions that BD-1.2 established dominance over BD-2. Legend could be more explicit as to what this figure represents.

Figure 3c – did all serotype switched strains harbour the same 110bp sequence insertion? What proportion of BD-1.2 strains harboured the change?

Other:

Line 148/9 - Define HPD.

REVIEWER COMMENTS

Reviewer #1 (Remarks to the Author):

The authors investigated the genomic characteristics of *V. cholerae* O1 responsible for the 2022 Dhaka cholera epidemic in a global phylogenetic context of 960 7PET strains from 88 countries. They found that the 2022 outbreak was caused by a subclade, BD-1.2 that was present in Bangladesh since 2016. Moreover, they claimed that this subclade successfully established dominance over the endemic BD-2 clade mainly through genetic changes including serotype switch in BD-1.2 from Ogawa to Inaba.

Reply: Thanks for the valuable suggestions and comments which were very helpful for us to improve our manuscript.

Specific comments

1. In the title add O1 to *Vibrio cholerae*

Reply: Added.

2. Since BD 1.2 has existed since 2016, it should not be referred to as a “new” subclade.

Reply: We agree, but this is a newly recognized subclade since no previous studies have reported the BD-1.2 from Bangladesh. We have referred the subclade as ‘newly recognized subclade’ in the revised version of our manuscript.

3. Since this manuscript might be coming out when the outbreak is over, I suggest that sentences such as “Bangladesh, currently is experiencing massive cholera” be revised to reflect this was in the recent past.

Reply: Actually, we prepared the draft manuscript at a time when the cholera outbreak was ongoing. Nonetheless, we have revised as suggested.

4. The study is essentially reporting on the recent 2022 outbreak, however, only 21 isolates from this outbreak were included in the analyses. That is quite a limitation to have 21 isolates while there were about 40,000 diarrhea cases within a month. This has to be addressed because it can have a significant effect on the conclusions of the study, especially about the subclade BD1.2 found in these 21 isolates, knowing that a larger strains collection for this huge outbreak could potentially cover and reveal more subclades...

Reply: Thanks for the important question. In fact, we focused on the source attributes of the cholera bacterium associated with the massive outbreak which broke out surpassing all past

records of the daily patients. We agree that more strains would be informative, but our resource is limited which allowed us to sequence 21 strains that we isolated during the peak of the massive outbreak.

A small number of strains could be instructive for identifying source attribute. For example, to identify source of the Haitian cholera outbreak only 23 *V. cholerae* strains were used that suggested the Asian origin for the bacterium (Chin et al. 2011, N Engl J Med, DOI: 10.1056/NEJMoa1012928). Follow-up molecular typing and genomic studies on more *V. cholerae* strains isolated in Haiti upheld the conclusions made earlier by Chin et al. (2011)(Hendriksen et al., 2011, mBio, DOI: 10.1128/mBio.00157-11; Reimer et al., 2011, DOI:10.3201/eid1711.110794). Likewise, a recent study sequenced 42 strains to reveal the source attributes of the outbreak strains in Yemen where over 1.1 million cases and 2,300 deaths were reported during 2016-2017 (Weill et al., Nature, 2019; DOI: 10.1038/s41586-018-0818-3).

In our study, we have defined BD-1.1 and BD-1.2 subclades based on their source attributes, not based only on the genetic distance. According to our data, BD-1.2 strains did not directly evolve from BD-1.1. Likewise, BD-1.1 did not directly evolve from BD-1. Our BEAST phylodynamic analysis showed both BD-1.1 and BD-1.2 strains had the most recent common ancestor with the strains isolated from neighboring countries, indicating different source attributes for the subclades. Although not impossible, to our knowledge, it is highly unlikely that strains belonging to other subclades were introduced from other countries into Bangladesh around the same time to cause cholera outbreak independently, as did the BD-1.2. Nonetheless, we do not rule out the possibility that many strains could evolve from the basal strains for the BD-1.2 and have contributed to the 2022 outbreak, but as per our definition, those strains belonged to BD-1.2 subclade. For example, the Ogawa and Inaba strains isolated during the outbreak clustered separately under the BD-1.2 subclade, however we did not define them as separate subclades because they evolved from a common ancestor belonging to the lineage BD-1.2 in Bangladesh. Again, strains isolated from Dhaka and Mathbaria (Dhaka is in the center and Mathbaria is in coastal area, and around 400 km apart from each other) between 2018 and 2019 clustered with BD-1.2, reflecting its predominance in Bangladesh in the recent years. In addition, other genetic markers such ctxB genotype (ctxB1 for BD-2, and ctxB7 for BD-1.2) and drug sensitivity (BD-1.2 sensitive and BD-2 resistant to Tetracycline) that we routinely test in our laboratory also suggest predominance of BD-1.2 like strains responsible for cholera in Bangladesh since 2018.

5. For information, it would be good if the authors provide in bracket the years of isolates of the 267 genome sequences added from their laboratory collections, as this helps to understand their epidemiological link to the current outbreak. For instance, Fig 2 C shows a population replacement/takeover by BD1.2. This interpretation is however based on just 21 isolates... in other words, we cannot rule out here that BD2 strains were not part of the 2022 outbreak.

Reply: We have added the years of isolation for the 267 genome sequences in the updated manuscript. All of the 21 sequenced strains belonging to the subclade BD-1.2 in our study suggests predominance of this lineage as an etiology of the 2022 massive outbreak, nonetheless, that does not rule out the possibility that BD-2 strains were part of the 2022 outbreak considering that we have analyzed only a limited number of the outbreak strains. Despite the number of outbreak strains analyzed were limited to 21, our results show predominance of the BD-1.2 strains, and a contributor of the 2022 massive cholera outbreak

in Dhaka, not the BD-2 as we did not find BD-2-like strains carrying ctxB1 genotype after 2019 in any of our cholera surveillance studies ongoing in our laboratory.

In 2018, a large cholera outbreak occurred in Dhaka; and between April 2 and May 12, icddr,b hospital treated 29,212 diarrheal patients (Hasan et al., 2021, PLOS Neglected Tropical Diseases; DOI: 10.1371/journal.pntd.0009953). In our study, we have included the whole genome sequence data of 44 strains isolated in 2018 in Bangladesh and found 31 strains shared cluster with BD-1.2, and only 13 strains shared cluster with the BD-2. This result suggests that BD-1.2 strains were predominantly associated with cholera outbreak in 2018. In subsequent years, we found only one strain isolated in 2019 shared cluster with the BD-2, while all of the 15 strains sequenced shared cluster with BD-1.2, suggesting that this was the predominant type in Bangladesh before 2022.

We showed BD-1.2 strains as part of the global clade, which have been associated with outbreaks globally, including the 2010 outbreak in Haiti, and the subsequent epidemic in Yemen, 2016-2022. In Bangladesh, BD-2 strains were associated with endemic cholera between 2004-2019, and predominantly during 2013 – 2017. But, we did not record a massive outbreak of the 2022 scale in that period, presumably, because of the lineage-specific immunity of the risk population. All these suggest recent clonal expansion of the BD-1.2 responsible for the massive outbreak by displacing the BD-2 strains that were predominant and responsible for endemic cholera in Bangladesh.

6. In fig 1a, the title/names of each ring are needed and should be readable as this will guide the reader to know which ring corresponds to which legend.

Reply: Titles/names of each ring were added as suggested.

7. There is a need for a better reference announcing the 2022 outbreak compared to the press release in Reference 1... it would be of interest to see if the outbreak has ended and how many total cumulative cases have been recorded now.

Reply: In the updated version of MS, we have incorporated the icddr,b official announcement of the 2022 outbreak that was posted on the official website of icddr,b and updated/corrected information accordingly.

8. I'm in doubt whether strain DMAVC-20 from Nov 21 was part of this outbreak (in Table S1, it is also described as "this study" in the "reference column"

Reply: We have not counted DMAVC-20 as an outbreak strain, although this strain was sequenced along with the outbreak strains in this study.

9. The extended data in figure 2c is complex to interpret and might be useful to add more legends

Reply: Additional legends to the Extended Data Fig. 2 were added in the updated MS.

10. The Yemen and Tanzanian T13 strains have ctxB7 and are Ogawa serotypes with

MLST 69 and reported to originate from Asia. However, the BD1.2 strains in this study which seem to be related to the African T13 strains are rather Inaba. Could the authors please confirm the MLST of these BD1.2? For information, most VC O1 Inaba tend to be of ST515 and often with a ctxB1, which is not the case here.

Reply: MLST analysis was performed using the schemes from PubMLST and found strains of both serotypes Inaba and Ogawa had ST-69. Phylodynamic analysis suggests BD-1.2 and T13 strains had common ancestry with the strains isolated from India. Here is the list of strains isolated during the massive outbreak and the corresponding ST-types and gene alleles.

Isolate	ST	New ST	Serotype	adk	gyrB	mdh	metE	pntA	purM	pyrC
BD_2022_DMAVC-1	69		Inaba	7	11	4	37	12	1	20
BD_2022_DMAVC-2	69		Inaba	7	11	4	37	12	1	20
BD_2022_DMAVC-3	69		Inaba	7	11	4	37	12	1	20
BD_2022_DMAVC-4	69		Ogawa	7	11	4	37	12	1	20
BD_2022_DMAVC-5	69		Inaba	7	11	4	37	12	1	20
BD_2022_DMAVC-6	69		Inaba	7	11	4	37	12	1	20
BD_2022_DMAVC-7	69		Inaba	7	11	4	37	12	1	20
BD_2022_DMAVC-8	69		Ogawa	7	11	4	37	12	1	20
BD_2022_DMAVC-9	69		Inaba	7	11	4	37	12	1	20
BD_2022_DMAVC-10	69		Inaba	7	11	4	37	12	1	20
BD_2022_DMAVC-11	69		Inaba	7	11	4	37	12	1	20
BD_2022_DMAVC-12	69		Inaba	7	11	4	37	12	1	20
BD_2022_DMAVC-13	69		Inaba	7	11	4	37	12	1	20
BD_2022_DMAVC-14	69		Inaba	7	11	4	37	12	1	20
BD_2022_DMAVC-15	69		Ogawa	7	11	4	37	12	1	20
BD_2022_DMAVC-16	69		Ogawa	7	11	4	37	12	1	20
BD_2022_DMAVC-17	69		Ogawa	7	11	4	37	12	1	20
BD_2022_DMAVC-18	69		Ogawa	7	11	4	37	12	1	20
BD_2022_DMAVC-19	69		Ogawa	7	11	4	37	12	1	20
BD_2022_DMAVC-21	69		Inaba	7	11	4	37	12	1	20
BD_2022_DMAVC-22	69		Inaba	7	11	4	37	12	1	20

11. Can the authors also please show the mutations in the wbeT region that are known to be responsible for the switch from Ogawa to Inaba (often Del_TGTAC at nt

position 24,28)? Because figure 3c from Artemis (about *rfbT*) is not informative enough on this.

Reply: We did not find this deletion (Del_TGTAC at nt position 24,28) for the switch from Ogawa to Inaba strains in our analysis. Gene sequences were aligned using MEGA11 and highlighted the region (nt position: 24-28) by green color and the insertion (nt position: 27-28 ins112bp) in alignment of *rfbT* genes of BD-1.2 Inaba and Ogawa strains. In the earlier version, we made a mistake in counting base-pair of the insertion region.

```

BD-1.2_Ogawa  ATGAAACATCTAATAAAAACTA TGTACAAAAATTAATTAACAAGAGCTTGATGCTATTTCAGTCAAAGTCTGTTTCATGATAATCGAAA
BD-1.2_Inaba  ATGAAACATCTAATAAAAACTA TGTACAAAAATTAATTAACAAGAGCTTGATGCTATTTCAGTCAAAGTCTGTTTCATGATAATCGAAA
BD-1.2_Ogawa  CTTCATTTACAATGGAGAGTTTTAATTCCTTGAAGCGAATTTGGATGGCATTGTTTTCCAGAGTGCAGTTGAACCATGCTTTAAGCT
BD-1.2_Inaba  CTTCATTTACAATGGAGAGTTTTAATTCCTTGAAGCGAATTTGGATGGCATTGTTTTCCAGAGTGCAGTTGAACCATGCTTTAAGCT
BD-1.2_Ogawa  ACAAAAACCCAAACTTTGATTTAGGTATGCGTCACTGGATTGTTAATCATTGTAAGCATGACACCCTTATATTGATATCGGTGCAAAC
BD-1.2_Inaba  ACAAAAACCCAAACTTTGATTTAGGTATGCGTCACTGGATTGTTAATCATTGTAAGCATGACACCCTTATATTGATATCGGTGCAAAC
BD-1.2_Ogawa  GTTGGAACTTTCTGTGGAATCGCTGCTCGTCATATTACACAAAGGAAAAATTTATAGCGATA-----
BD-1.2_Inaba  GTTGGAACTTTCTGTGGAATCGCTGCTCGTCATATTACACAAAGGAAAAATTTATAGCGATA TGTAGTGGCATAGTGAATTTGGTCACTTA
BD-1.2_Ogawa  -----GAACCA
BD-1.2_Inaba  TTTAGAGGTGATATCATCACCTCATAACGATTATTTGGGAAAACCTTAAAACCGTGGCCAATTTAGTTGACCACTACAATTTGAACCA
BD-1.2_Ogawa  CTCACAGAAATGGAAAATAGTATTAGGATGAATGTTCAATTAATAATCCACTAGTTGAGTTTCATCATTTTGGCTGTGCAATAGGTGA
BD-1.2_Inaba  CTCACAGAAATGGAAAATAGTATTAGGATGAATGTTCAATTAATAATCCACTAGTTGAGTTTCATCATTTTGGCTGTGCAATAGGTGA
BD-1.2_Ogawa  GAATGAAGGGGAAAAATATTTTCGAAGTTTATGAGTTTGATAATAGGGTGTGCATCATTATATTTTCAAAAAATACAGACATAGCAGATA
BD-1.2_Inaba  GAATGAAGGGGAAAAATATTTTCGAAGTTTATGAGTTTGATAATAGGGTGTGCATCATTATATTTTCAAAAAATACAGACATAGCAGATA
BD-1.2_Ogawa  AGGTTAAAAATAGCCAAGTTCGGTTAGAAAGTTAAGTAGTTTAGATATATCGCCTACTAACTCTGTAGTTATAAAAAATTGATGCTGAA
BD-1.2_Inaba  AGGTTAAAAATAGCCAAGTTCGGTTAGAAAGTTAAGTAGTTTAGATATATCGCCTACTAACTCTGTAGTTATAAAAAATTGATGCTGAA
BD-1.2_Ogawa  GGCGCAGAAATAGAGATATTAACCAGATTTACGAATTCACAGAAAAGCATAATGGAATTGAATATTATATTGCTTTGAATTTGCAAT
BD-1.2_Inaba  GGCGCAGAAATAGAGATATTAACCAGATTTACGAATTCACAGAAAAGCATAATGGAATTGAATATTATATTGCTTTGAATTTGCAAT
BD-1.2_Ogawa  GGGTCATATACAGAGGTCTAATAGAACTTTTGATGAGATTTTAAACATAATAAACTCAAATTCGGAAGTAAGGCATATTTTATTTCATC
BD-1.2_Inaba  GGGTCATATACAGAGGTCTAATAGAACTTTTGATGAGATTTTAAACATAATAAACTCAAATTCGGAAGTAAGGCATATTTTATTTCATC
BD-1.2_Ogawa  CATTATCATCCGCTGAACATCCTGAGTTTAATAAAGCAACGAGGATATTAAATGGGAATATCTGTTTTAAATATGTATCATAA
BD-1.2_Inaba  CATTATCATCCGCTGAACATCCTGAGTTTAATAAAGCAACGAGGATATTAAATGGGAATATCTGTTTTAAATATGTATCATAA

```

12. It would be important to know and compare how was the ICE SXT in the BD-1.2 strains when describing the AMR data. Similarly, changes in the quinolone resistance genes *parC* and *gyrA* are important markers in the resistance epidemiology of *V. cholerae*.

Reply: Nucleotide blast was used to match contigs of BD-1.2 strains with seven publicly available sequences of the Integrative and conjugative elements (ICEs)- ICEVchban5 (GQ463140.1), ICEVchind4 (GQ463141.1), ICEVchind5 (GQ463142.1), ICEVchmex1 (GQ463143.1), ICEVflInd1 (GQ463144.1), ICE^{TET} (MK165649.1), and ICE^{GEN} (MK165650.1). All strains of BD-1.2 blast search yielded high bit scores when aligned with ICE^{GEN} (MK165650.1), ICEVchInd5 (GQ463142.1), or ICEVchBan5 (GQ463140.1), suggesting genetic similarity (Supplementary Data 3). In addition, BD-1.2 strains have Haitian *gyrA* gene allele (Ser 83 Ile) and mutant *ParC* (ser85Ile substitution).

13. The method is well described and the study well conducted, a suggestion is that RaxML generates better probability compared to IQ-Tree.

Reply: We used IQ-Tree, IQ-TREE has a model finder to determine reconstruction parameters. We conducted 1000 bootstrap sampling that helps to construct a robust phylogenetic tree.

14. As one of the key take-home messages from this paper is the takeover of BD1.2, a table included directly in the manuscript showing characteristic differences between BD1.2 and BD2 should be added.

Reply: We have shown the genetic differences between BD-1 and BD-2 in our recent publication (Monir et al., 2022; Microbiology Spectrum). Nonetheless, as per suggestion, we have included a table showing the major differences between BD1.2 and BD2 (Table 1).

15. If possible from archives, authors are encouraged to enrich the tree with more strains from the 2022 outbreak to increase the power of the conclusions.

Reply: Analysis of more genomes might be useful, nonetheless, utilizing the sequenced data of 21 strains, we have been able to demonstrate that a newly recognized lineage, BD-1.2, took over by displacing the Asian lineage strains, BD-2, in Bangladesh and was predominantly associated with massive cholera outbreak 2022 that broke all past records of daily patients admitted at icddr,b .

Reviewer #2 (Remarks to the Author):

This manuscript reports on the genetic characteristics of the 2022 cholera outbreak in Bangladesh, based on sequences from 21 clinical isolates. It compares these genomes with ~960 sequences retrieved from public databases as well as from their laboratory collection. The data indicate that the strains belonged to a separate subclade of BD-1-like strains, which represented a switch from the previously dominant BD-2 lineage. The data presented here are timely and noteworthy, and once again, demonstrates how genomic microbial surveillance of clinically sourced isolates can provide important insights into the spread of disease and evolution of virulence at local and global levels.

The manuscript contains appropriate descriptions of the whole genome sequence data and the comparative analyses performed are consistent with the methodology used in this field.

Comments for the authors:

1. The introduction to the manuscript is somewhat challenging to follow for a non-Vibrio expert:

a. Line 72 (and elsewhere e.g. line 81)– these are Vibrio specific terms that should be defined and their importance explained.

Reply: As suggested, we have defined VSP-I, VSP-II, SXT ICE, and VPI-1, and also described the functions.

b. Line 77-79+ - As written it is not clear if the lineages mentioned in the first sentence are the same as or distinct from those mentioned in the second sentence. The manuscript needs a better introduction to global Vibrio phylogeny and then the local situation in Bangladesh.

Reply: We have included description about global vibrio phylogeny in the revised version.

c. Line 89 – poor phrasing “It was concluded imperative”

Reply: Revised, as suggested.

2. Number of strains sequenced – the authors sequenced quite a small number of isolates (n=21) in relation to the outbreak size (~40,000 cases a month) and when compared to the numbers sequenced in other (similar) cholera papers. Why was this number chosen and how were these selected from the huge numbers of patients admitted to hospital (e.g. patient or disease characteristics?). Is this number sufficient to describe the outbreak agent?

Reply:

(Answer to comments: “Number of strains sequenced – the authors sequenced quite a small number of isolates (n=21) in relation to the outbreak size (~40,000 cases a month) and when compared to the numbers sequenced in other (similar) cholera papers.”)

Thanks for the important question. Other cholera studies sequenced similar number of strains. For example, to identify source of the massive Haitian cholera outbreak in 2010, the Whole Genome of only 23 outbreak strains of *V. cholerae* were sequenced (Chin et al. 2011, N Engl J Med, DOI: 10.1056/NEJMoa1012928). Likewise, a recent study sequenced 42 strains to reveal the source attributes of the outbreaks in Yemen where over 1.1 million cases and 2,300 deaths were reported during 2016-2017 (Weill et al., Nature, 2019; DOI: 10.1038/s41586-018-0818-3).

(Answer to comments: “Why was this number chosen and how were these selected from the huge numbers of patients admitted to hospital (e.g. patient or disease characteristics?)”

In this study, we focused on the source attributes of the cholera bacterium associated with the massive 2022 outbreak which broke out surpassing all past records. To our knowledge, there has been no statistical approach for choosing appropriate number of strains for comparative genomic studies to understand the causal agent and the source attributes. As mentioned above, we strongly believe we have sequenced a reasonable number of strains as did the previous cholera studies. We took stool samples at random from hospitalized individuals who had tested positive for cholera by rapid dipstick test and culturing methods.

(Answer to comments: “Is this number sufficient to describe the outbreak agent?”

Our answer is yes, as our results showed explicitly. We have a couple of examples in the field, such as the comparative genomic analysis of only 23 outbreak strains of the Haitian cholera outbreak showed Asian origin of *V. cholerae* O1 (Chin et al. 2011, N Engl J Med, DOI: 10.1056/NEJMoa1012928). Further genomic studies support the same conclusions made by Chin et al. 2011 for the origin of Haitian cholera (Hendriksen et al., 2011, mBio, DOI: 10.1128/mBio.00157-11; Reimer et al., 2011, DOI:10.3201/eid1711.110794). Our cholera research team at icddr,b has been involved in collecting stool samples from patients, and so far, we have only found *V. cholerae* O1 strains possessing the ctxB7 genotype as of our lab results, which suggest a population take over for the BD-1.2 strains by displacing BD-2 strains responsible for endemic cholera in the preceding years in Bangladesh (as shown in Figure 2c).

As for the etiologies, we have seen in the past, not all of the patients admitted to our icddr,b hospital proved cholera, but as per our hospital data *V. cholerae* is the major etiology for the 2022 diarrhea outbreak (Data unpublished, will be published soon by hospital team of icddr,b). In this study, we characterized genomic and source attributes of the major etiology of the diarrhea outbreak.

3. Line 294-6 – check this sentence as it does not make sense to me “...yet victims infected with either serotype generally not develop clinically apparent disease”. Aren’t strains all taken from hospitalised patients?

Reply: We discussed about the non-symptomatic cases, who had been infected by different serotypes in before. However, in this study we only included symptomatic cases. Therefore, we have changed the sentence in the revised version.

Figure related queries:

Figure 1 – what is significance of including/distinguishing Dhaka vs Mathbaria locations for the phylogenetic tree?

Reply: Dhaka is in the center and Mathbaria is in coastal area, and around 400 km apart from each other. Strains isolated from Dhaka and Mathbaria between 2018 and 2019 clustered with BD-1.2, which reflect predominance of the strains in Bangladesh in the recent years.

Figure 2c – this is a nice pictorial representation of the strain changes over time. Is this data specific to strains isolated only in Dhaka? If so, the image infers that between 10-50% were found to belong to BD-1.2 in 2022 - where do the remaining strains situate in terms of their genetics? How does this fit with the authors conclusions that BD-1.2 established dominance over BD-2. Legend could be more explicit as to what this figure represents.

Reply: All strains belonging to BD-1, BD-1.1, BD-1.2, and BD-2 isolated in Bangladesh including Dhaka and Mathbaria were counted (100%). Meanwhile, the bubble size is weighted by number of total Bangladesh strains belong to all of these groups. For example- total 286 strains isolated between 1999 and 2022 belonged to any of the lineages: BD-1, BD-1.1, BD-1.2, and BD-2. The number of sequenced strains included 21 from the 2022 outbreak. Therefore, the relative frequency of the BD-1.2 strains is $(21/286 = 7\%)$. This is the basis that, all of the strains isolated in 2022 belonged to BD-1.2, which are 7% of the total Bangladeshi strains isolated between 1999 to 2022.

Figure 3c – did all serotype switched strains harbour the same 110bp sequence insertion? What proportion of BD-1.2 strains harboured the change?

Reply: Within 21 strains isolated in 2022, 7 were Ogawa and 14 Inaba. All of the Inaba strains had this insertion (2 strains had 113bp insertion and remaining 12 had 112bp insertion). We made mistake in counting the nt's, however, corrected in the updated version.

Other:

Line 148/9 - Define HPD.

Reply: Defined.

** See Nature Portfolio's author and referees' website at www.nature.com/authors for information about policies, services and author benefits.

This email has been sent through the Springer Nature Tracking System NY-610A-NPG&MTS

Confidentiality Statement:

This e-mail is confidential and subject to copyright. Any unauthorised use or disclosure of its contents is prohibited. If you have received this email in error please notify our Manuscript Tracking System Helpdesk team at <http://platformsupport.nature.com> .

Details of the confidentiality and pre-publicity policy may be found here <http://www.nature.com/authors/policies/confidentiality.html>

Privacy Policy | Update Profile

DISCLAIMER: This e-mail is confidential and should not be used by anyone who is not the original intended recipient. If you have received this e-mail in error please inform the sender and delete it from your mailbox or any other storage mechanism. Springer Nature Limited does not accept liability for any statements made which are clearly the sender's own and not expressly made on behalf of Springer Nature Ltd or one of their agents.

Please note that Springer Nature Limited and their agents and affiliates do not accept any responsibility for viruses or malware that may be contained in this e-mail or its attachments and it is your responsibility to scan the e-mail and attachments (if any).

Springer Nature Ltd. Registered office: The Campus, 4 Crinan Street, London, N1 9XW. Registered Number: 00785998 England.

REVIEWERS' COMMENTS

Reviewer #1 (Remarks to the Author):

Dear Authors,

Thank you for the great details in your review responses and for revising the manuscript.

Regarding my main concern about the number of sequenced isolates (which I have now noticed was raised by other reviewers as well), I must admit that I like the arguments, justifications, and examples provided by the authors.

I however still think that for a massive outbreak like the one investigated in this study, including more samples from different stages of the outbreak (until the end) from different areas will be more representative and supportive of the conclusions... And it will make the study stronger I believe for a journal like Nat Com.

The editors can decide on that.

The revised manuscript is a good read and it was an honor revising this work. I look forward to printing the final version when it gets published.

Best regards

REVIEWERS' COMMENTS

Reviewer #1 (Remarks to the Author):

Dear Authors,

Thank you for the great details in your review responses and for revising the manuscript.

Regarding my main concern about the number of sequenced isolates (which I have now noticed was raised by other reviewers as well), I must admit that I like the arguments, justifications, and examples provided by the authors.

I however still think that for a massive outbreak like the one investigated in this study, including more samples from different stages of the outbreak (until the end) from different areas will be more representative and supportive of the conclusions... And it will make the study stronger I believe for a journal like Nat Com.

The editors can decide on that.

The revised manuscript is a good read and it was an honor revising this work. I look forward to printing the final version when it gets published.

Best regards

Reply:

Thanks for the positive feedback. As we mentioned in the previous reply, the analysis might benefit from including more genomes, however utilizing the sequenced strains, we were able to demonstrate that *Vibrio cholerae* O1 strains belong to BD-1.2 recently emerged in Bangladesh and linked to 2022 massive cholera outbreak.

Although it is currently not possible for us sequencing additional strains from archives due to our funding constraints, however we tested additional *V. cholerae* by randomly selecting 30 strains isolated between March and September, 2022, in order to support our study findings and strengthen the conclusions made. The *ctxB* genotype was determined by a double-mismatch-amplification mutation assay (DMAMA) PCR and drug response patterns using disk diffusion assays. According to our results, all of the tested strains carried *ctxB7*, and proved tetracycline sensitive as observed for the BD-1.2, which appeared in sharp contrary to the BD-2 carrying *ctxB1*, and tetracycline resistance as markers, supporting the overall findings of the present study.